# The Variation of Counter-Electrojet Current at the Southeast Asian Sector during Different Solar Activity Levels



Nur Izzati Mohd Rosli [1], Nurul Shazana Abdul Hamid [1,2,*], Mardina Abdullah [2,3], Khairul Adib Yusof [2,4], Akimasa Yoshikawa [5], Teiji Uozumi [5] and Babatunde Rabiu [6]

[1] Department of Applied Physics, Faculty of Science and Technology, Universiti Kebangsaan Malaysia, Bangi 43600, Malaysia; p98980@siswa.ukm.edu.my
[2] Space Science Centre (ANGKASA), Institute of Climate Change, Universiti Kebangsaan Malaysia, Bangi 43600, Malaysia; mardina@ukm.edu.my (M.A.); adib.yusof@upm.edu.my (K.A.Y.)
[3] Department of Electrical, Electronic and Systems Engineering, Faculty of Engineering and Built Environment, Universiti Kebangsaan Malaysia, Bangi 43600, Malaysia
[4] Department of Physics, Faculty of Science, Universiti Putra Malaysia, Serdang 43400, Malaysia
[5] International Center for Space Weather Science and Education, Kyushu University, Fukuoka 812-8581, Japan; yoshikawa.akimasa.254@m.kyushu-u.ac.jp (A.Y.); uozumi@serc.kyushu-u.ac.jp (T.U.)
[6] Centre for Atmospheric Research, National Space Research and Development Agency NASRDA, Kogi State University Campus, Anyigba 272102, Nigeria; tunderabiu2@gmail.com
* Correspondence: zana@ukm.edu.my; Tel.: +60-389-215-914

**Abstract:** Studies on counter-electrojet currents (CEJ) using ground data revealed that this current could occur simultaneously among locations that are less than 30° longitude apart. In our work, the symmetricity of CEJ variation between the west and east of Southeast Asia, separated by ~25°, was preliminarily examined according to its types: morning (MCEJ) and afternoon (ACEJ). Since most of the past studies had overlooked the occurrence after dusk, the monitoring period was also extended from 18:00 to 21:00 LT, namely, the post-sunset depletion (PSD). The magnetometer station in Davao, Philippines (DAV) and Langkawi, Malaysia (LKW) were chosen to represent the east and west parts. The EEJ index (i.e., EUEL) over the periods of the solar cycle 24 (2008–2018) was utilized specifically during magnetically quiet days (Kp < 3). As the result, both parts symmetrically showed that MCEJ and ACEJ were positively and negatively correlated with the F10.7 index. Contrarily, MCEJ and ACEJ were asymmetrically prominent in the east and west. CEJ types also varied symmetrically with the season, especially for MCEJ and ACEJ (at high level), prominent during Equinox and J-solstice. Post-sunset depletion (PSD) in both parts was symmetrically solar activity independent, as no correlation with the F10.7 index was observed in the extended observation. PSD that varied symmetrically with season was also solar activity independent, except in the east during Equinox, where it was negatively correlated with the F10.7 index. Our finding also revealed that PSD was prominent during Equinox, except for the high level in the west part.

**Keywords:** CEJ; solar activity level; season; Southeast Asia; solar cycle 24

## 1. Introduction

The presence of high electron density and electrodynamic processes in the Earth's ionospheric layer resulted in the development of current systems. EEJ current exists within $\pm 3°$ dip equator in the E region of the ionosphere and forms a narrow band-like current with an eastward direction at the equator. EEJ current can be monitored through the daytime variation of ground equipment's horizontal geomagnetic field component (H-component) [1,2]. Occasionally, the H-component is temporarily depressed below the night-time level as the current flow reverses to the west. This reversal event, which is known as counter-electrojet (CEJ) current, typically lasts 4 to 5 h and is the main focus of the present work [3]. CEJ is usually caused by environmental factors dependent on

geomagnetic conditions. For example, CEJ during quiet days could be driven by changes in the atmospheric tides that dominate the global wind system at ionospheric heights [4,5]. The variability for both CEJ and EEJ events are dependent on the variation in local time, longitude, seasonal dependence, lunar cycles, magnetic activity, and solar activity [6–12].

The CEJ study has been gradually expanded to a global scale through the integration of ground, satellite, and ionospheric measurements. Over a large longitudinal separation range, the variability of CEJ showed that morning occurrence dominated more than afternoon globally, except for the Indian sector. The highest occurrence rate was recorded in the Brazilian sector than elsewhere [10]. CEJ occurrence might not take place simultaneously on the same day if the longitude separation was less than 30° [13–15], but a study in the Indian sector revealed more simultaneous occurrences between ~15° separations after 1500 LT [16]. On the other hand, a study on the EEJ-Sq (solar quiet) relationship demonstrated an uncommon behavior in Southeast Asia when compared to India and America, due to the disturbance in dynamo regions [17]. These aforementioned works motivate and present an opportunity to preliminarily examine the symmetricity of CEJ variation between the insular (east) and peninsular (west) of Southeast Asia that are separated within 30° longitude before a comprehensive study can be conducted as a continuation thereof. Moreover, detailed studies of CEJ, especially in the western part of Southeast Asia, are limited as only data from the east were primarily utilized in past studies [8,10,18].

Although it is widely accepted that CEJ must occur between 06:00 and 18:00 LT, the determination of the local time interval is not generally agreed upon as it usually depends on the study focus, region, and CEJ occurrences themselves. For instance, an initial study that introduced the local time interval used the following classification for African and Indian sectors: morning (06:00–08:00 LT), noon (11:00–12:00 LT), afternoon (13:00–14:00 LT), and evening CEJ (15:00–18:00 LT) [19]. A recent study proposed a more extended period as it covered a more comprehensive longitudinal range that includes South American, African, Indian, and Southeast Asian sectors: morning (06:00–10:00 LT) and evening (14:00–18:00 LT) [10]. The disagreement over CEJ classification will complicate subsequent studies to perform fair comparisons between findings obtained in different sectors and at different times. Therefore, a suitable interval for CEJ observations in the Southeast Asian sector was applied based on its occurrences.

Other than that, there was also a possibility of CEJ occurrence after dusk, which most studies had overlooked since it happened during off-daytime. This uncommon event was once indicated by the interplanetary magnetic field (IMF) Bz movement [15]. Fundamentally, this finding contradicts the CEJ theory, which established CEJ as a daytime phenomenon, necessitating further exploration. Hence, the current study aims to examine CEJ dependency on solar activity and seasonal aspects, besides determining the symmetry of CEJ occurrence using data from the insular (east) and peninsular (west) of Southeast Asia that were separated by ~25° longitude. The monitoring period was then extended after dusk (18:00–21:00 LT) and introduced as post-sunset depletion (PSD).

## 2. Materials and Methods

This study used data from ground-based Magnetic Data Acquisition System (MAGDAS) magnetometer stations in Davao (DAV, GM Lat.: −2.22°, GM Lon.:197.90°) and Langkawi (LKW, GM Lat.: −3.30°, GM Lon.:172.44°) in the west and east of the Southeast Asian, respectively (refer Figure 1). In data processing, a series of EUEL index derivations on one-minute resolution data from 2008 to 2018 was performed as a representative analysis of the entire solar cycle 24. This index also referred to as the EEJ index was derived from a series of calculations, starting with the subtraction of the raw data with the median value of the northward geomagnetic component, H. The result gave the ERs value for each equatorial station, which was used to obtain another index, namely EDst. EDst, which served as the global magnetic variation indicator, was also the mean of ERs during the nighttime (LT = 18:00–06:00) from all available stations along the magnetic equator region.

EUEL was the local index which is the final outcome after ER was subtracted with the EDst index [20,21].

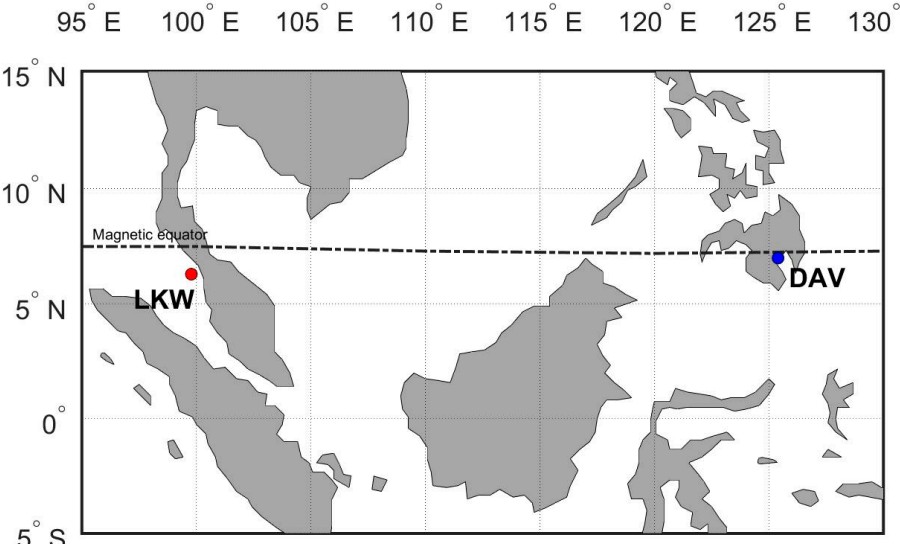

**Figure 1.** Location of the ground magnetometer stations used in the study: DAV (Davao) and LKW (Langkawi).

To avoid solar–terrestrial disturbances such as geomagnetic storms, only geomagnetically quiet days were considered in the analysis by referring to the global geomagnetic activity index, which is Kp-index at lower values (i.e., Kp ≤ 3).

Over the course of 11 years, two levels of solar activity were introduced: low (F10.7 ave = 75 s.f.u.)—2008, 2009, 2010, 2016, 2017, 2018; high (F10.7 ave = 125 s.f.u.)—2011, 2012, 2013, 2014, 2015. However, intermittent data from the LKW station from 2008 to 2010 were deemed insufficient to cover the whole year. Hence, the three consecutive years were excluded from the analysis for this station, leaving only the data from 2016, 2017, and 2018 in the low solar activity level category.

Following that, a set of criteria was established to ensure the accuracy of CEJ identifications. Based on daily preliminary inspections, it was determined that a CEJ could be identified when the EUEL value fell below—5 nT for at least 2 h, adapted from past studies [22]. The duration criterion was chosen to minimize false detections, since transient geomagnetic field depletion could occur from time to time due to other factors, such as instrument failures. In addition, depletion periods with any data gaps were excluded from the analysis to provide continuous CEJ.

Subsequently, CEJ that had been successfully identified were sorted based on several local time intervals. Since no occurrence was detected at noon (11:00–13:00 LT), the intervals proposed in this study were morning CEJ (MCEJ, 06:00–11:00 LT), and afternoon CEJ (ACEJ, 13:00–18:00 LT). PSDs were identified between 18:00 LT and 21:00 LT to consider possible night-time occurrences. Figure 2a depicts an observation of MCEJ and ACEJ, while Figure 2b shows an example of PSD. The number of observation days collected from Davao and Langkawi stations year by year was illustrated in Figure 3 and the percentage of all occurrences was calculated according to their types using the following formula [8]:

$$\text{Occurrence \%} = \frac{\text{No. of occurrence}}{\text{No. of observation days}} \times 100\%$$

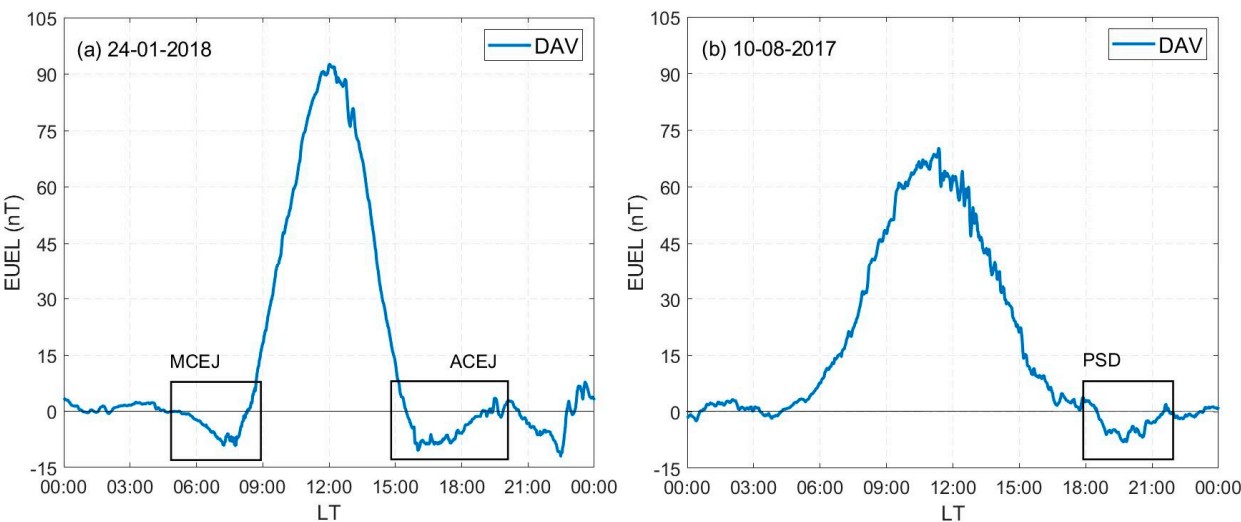

**Figure 2.** Examples include (**a**) MCEJ and ACEJ at DAV on 24 January 2018, and (**b**) PSD at DAV on 10 August 2017.

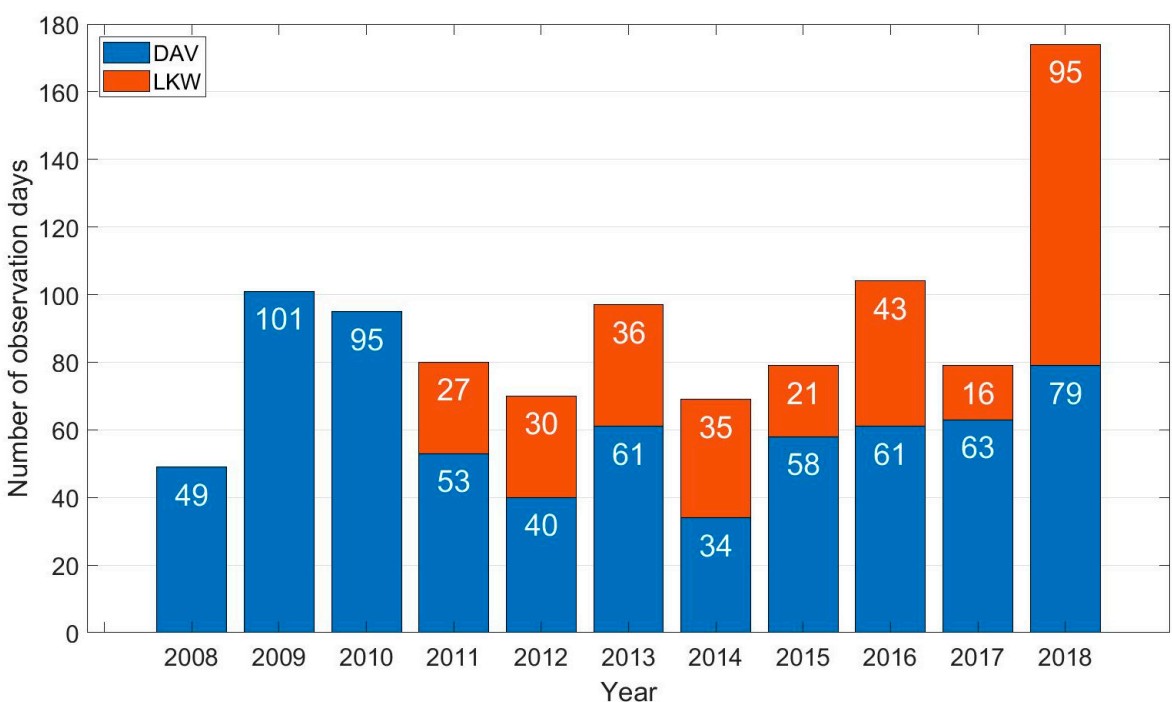

**Figure 3.** Number of observation days for 11 years of solar cycle 24 from different stations.

## 3. Results and Discussion

### 3.1. CEJ Occurrences Based on Local Times, Solar Activity Levels, and Seasons

The CEJ analysis results presented in this section were described according to local times, solar activity levels, and seasons. In Figure 4a,b, the occurrence percentages at DAV, which represented the east, were plotted alongside the solar flux index (F10.7) for MCEJ and ACEJ, respectively. Meanwhile, the pie charts in (c) and (d) show each CEJ type's occurrence percentages for low and high solar activity. Information in (e)–(h) is similar to (a)–(d) but for the LKW station which represents the west.

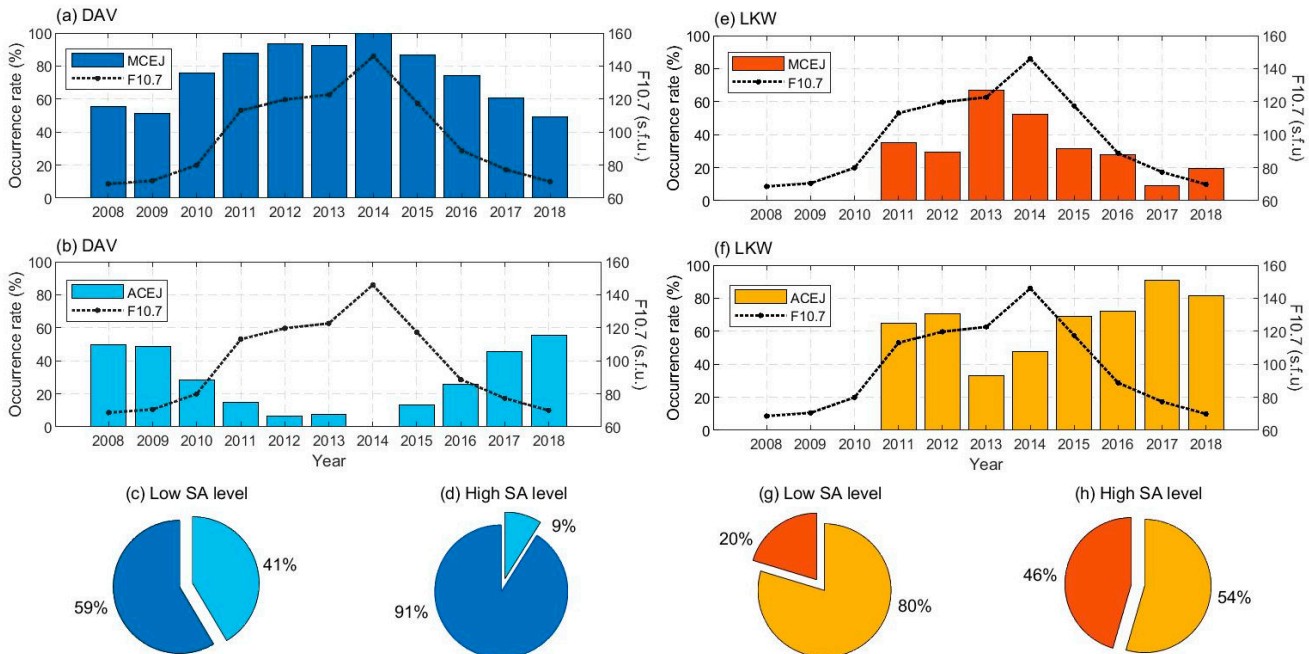

**Figure 4.** Percentage occurrences (left y-axis) of (**a**) MCEJ and (**b**) ACEJ at east (DAV) alongside F10.7 index (right y-axis) throughout solar cycle 24, as well as the overall percentages of both types of CEJ categorized into (**c**) low and (**d**) high solar activity levels. Information in (**e**–**h**) is similar to (**a**–**d**) but for the west (LKW).

Considering the years and solar activity levels, MCEJ and ACEJ in the east accounted for 70%, and 30% of the total detected CEJ. It was apparent that the prominent type of CEJ (the MCEJ) correlated well with F10.7 index values and occurred frequently (80–100%) during higher values (i.e., 2011–2014) compared to lower values (i.e., 2008–2010, 2016–2018). The opposite was observed for ACEJ, where the percentage dropped from 55% during lower index values to less than 20% during higher index values. The dependencies were quantified, supported by calculating the correlation coefficient, R, with R $\geq$ 0.6 being strongly correlated and R < 0.6 being weakly correlated [23]. MCEJ and ACEJ were positively (R = 0.95) and negatively (R = −0.95) correlated to the F10.7 index. Figure 4c,d shows that the dominant MCEJ increased from 59% during low to 91% during high solar activity levels.

CEJ dependency on the F10.7 index in the western part was quite similar to the east. According to Figure 4e,f, MCEJ occurred more frequently (30–70%) during the higher index than during the lower index (<40%), which is consistent with the observations in the east. In contrast, the ACEJ percentage reduced slightly from 90% during the lower index to less than 70% during the higher index. Correlations with the F10.7 index were found to be weaker compared to the east, with R = 0.77 for MCEJ and R = −0.78 for ACEJ, indicating moderate correlations. As afternoon type dominated 80% of CEJ events in the west, more ACEJs than MCEJs were observed for both solar activity levels, as illustrated in Figure 4g,h.

An outstanding correlation obtained in the presented results indicated that CEJ was highly dependent on solar activity. This is the first time such observation was reported for the Southeast Asian sector after observations during solar cycle 19 and 20 years in prior investigations could not be ascertained due to insufficient data and a small number of events [24]. Additionally, a novel finding and noteworthy contribution of the present work was the east–west asymmetrical behavior: the morning type's dominance in the eastern part, DAV (consistent with a previous observation [10]), and the afternoon type's dominance in the west, LKW. This asymmetricity might be induced by the EEJ of varying strengths at different locations as CEJ tend to be inhibited if strong EEJ occur [25]. More convincingly, a recent study in the same region had also discovered stronger EEJ at LKW

than at DAV, regardless of solar activity levels [23]. For that reason, a more detailed case study regarding the EEJ strength influence on CEJ types can be performed as a continuation of the current work to confirm the assumption.

Meanwhile, in the African sector, such asymmetry was attributed to differences in meridional currents across different longitudes [8] as the direction of the meridional current varies significantly with longitude, local time, and season [26]. However, this might have less relevance in corroborating our results since we adopted data from the insular and peninsular regions of Southeast Asia. Meanwhile, in the Indian sector [16], the divergence in CEJ occurrence between the mainland and insular regions, which are separated by ~15°, was related to local effects associated with eastward zonal winds influence [27], gravity wave atmospheric tidal interactions [28,29], and returning of westward currents due to atmospheric tidal effects [30,31]. Although these aforementioned mechanisms could possibly explain the asymmetry seen in the current work, a detailed investigation into its relevance, particularly in Southeast Asia, is required in a separate study.

Figures 5 and 6 present the seasonal variation of CEJ types in the east and west, respectively. Each figure was complemented with its annual rate that was estimated according to season—Equinox (March–May, September–November), June solstice (June–August), and December solstice (December–February) [32,33] in subplots (a) and (d). Also included are a series of pie charts to represent season dominancy at different solar activity levels. The results shown in these figures were supported with a monthly rate plot in Figure 7.

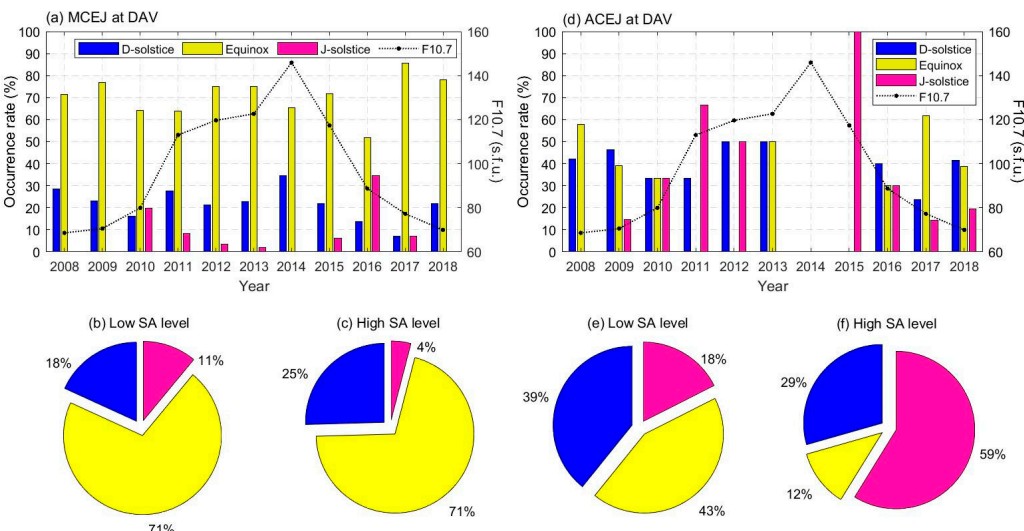

**Figure 5.** Seasonal variation (left y-axis) of (**a**) MCEJ and (**d**) ACEJ in the east (DAV) alongside F10.7 index (right y-axis), throughout the solar cycle 24, accompanied with its dominance at (**b**,**e**) low and (**c**,**f**) high solar activity levels.

In Figure 5a, the percentage of MCEJ in the east was prominent during Equinox. The seasonal pattern was anticorrelated with F10.7 index as poor correlation value was obtained. Equinox accounted for 71% of MCEJ occurrences at both levels from the pie charts (b) and (c), with D-solstice and J-solstice accounting for 29%. Similarly, the ACEJ seasonal pattern in (d) disagreed with the F10.7 index, associated with zero occurrences at the highest F10.7 index (2014). At a low level (refer to (e)), the percentage accounting for Equinox was counterpart with D-solstice and remained 18% for J-solstice, which contradicted the high solar activity level in (f) wherein 59% of the occurrence was dominated by J-solstice. The dominancy of this ACEJ during J-solstice, together with the prominent occurrence of MCEJ during Equinox seasons for high solar activity level, can be clearly observed in the CEJ monthly rate plot, as presented in Figure 7.

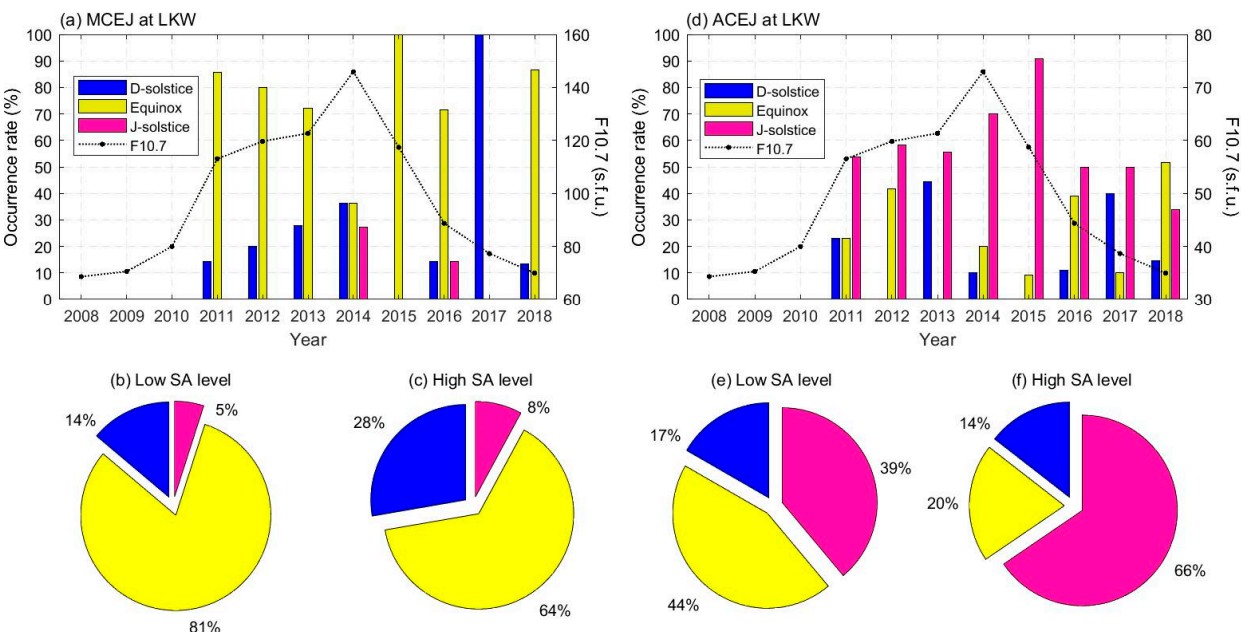

**Figure 6.** Seasonal variation (left *y*-axis) of (**a**) MCEJ and (**d**) ACEJ in the west (LKW) alongside F10.7 index (right *y*-axis), throughout the solar cycle 24, accompanied with its dominance at (**b**,**e**) low and (**c**,**f**) high solar activity levels.

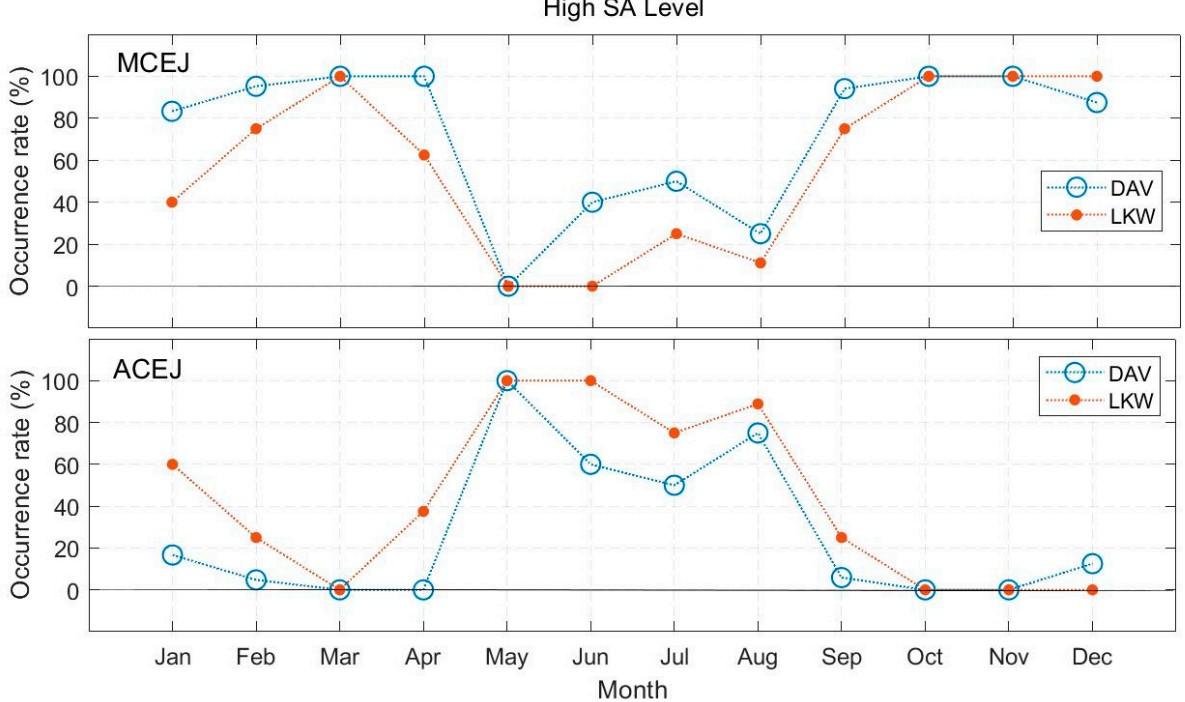

**Figure 7.** Monthly rate of MCEJ and ACEJ at during high solar activity level in the east (DAV) and west (LKW) sides.

Figure 6 presents results from LKW station, where a comparable percentage of MCEJ between the west and east was obtained in (a), regardless of seasons. A slight ACEJ correlation between occurrence during J-solstice season and the F10.7 index (R = 0.67) could be seen in (d). At different solar activity levels, the ACEJ percentage accounting for Equinox in (e) was initially equivalent to J-solstice, before dominating the high level in (f). Similarly

to the east, this prominent observation of ACEJ during J-solstice was confirmed by the CEJ monthly rate, as illustrated in Figure 7.

It was evident that the pronounced seasonal variation of CEJ at different solar activity levels exhibited east–west symmetrical behavior during Equinox season for the greatest MCEJ occurrence and J-solstice at high solar activity level for ACEJ. These presented results complemented recent findings in the Philippines [10] that the percentage of 11 years of solar cycle 24 rates peaked around Equinoctial months in morning observation and June in afternoon, regardless of the differences in levels of solar activity. At the same time, they reinforced their works, confirming that seasonal variability was mainly addressed by atmospheric tides and stationary planetary waves. ACEJ at a low solar activity level was excluded due to the corresponding percentage between Equinox and D-solstice in the east, and between Equinox and J-solstice in the west, in which we hypothesized was caused by insufficient data during low solar activity years at LKW, as represented by the three-year observation period in this study (2016–2018).

### 3.2. Post Sunset Depletion Occurrence (18:00 LT to 21:00 LT)

EUEL depletions below the night-time value that fulfills all predetermined CEJ criteria were detected between 18:00 LT and 21:00 LT, as illustrated in Figure 8a, where PSD occurred earlier at LKW station. PSD accounted for 28% of total EUEL depletions in the east (DAV) and 23% in the west (LKW). Overall, both parts were solar activity independent in a balanced way. As illustrated in Figure 8b, there was no correlation with the F10.7 index, as indicated by the low value of R = −0.14. Meanwhile, there was no significant difference in the percentage value between low and high solar activity levels in pie charts (c) and (d). Similarly, the correlation in Figure 8e was also weak, with R = −0.24, and both solar activity levels in (f) and (g) had no significant difference.

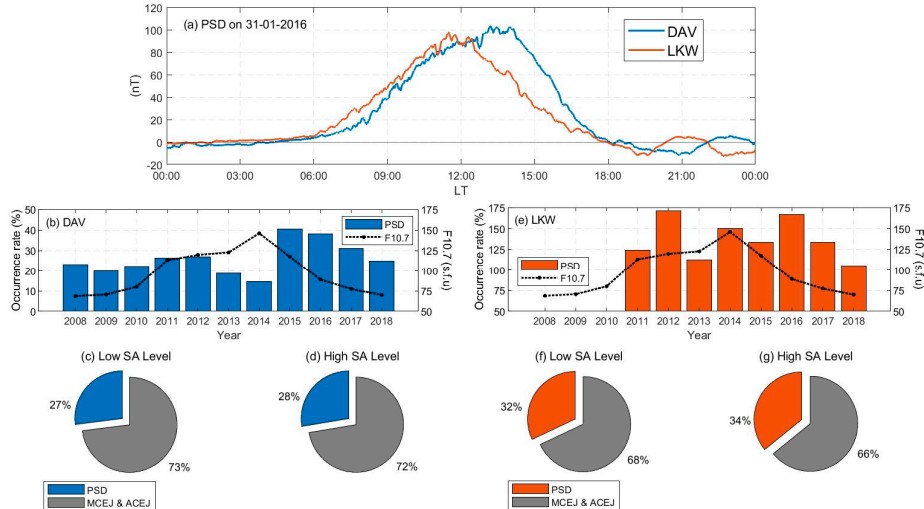

**Figure 8.** (**a**) PSD recorded on 1 January 2016, in both parts of Southeast Asia, (**b**) percentage occurrence of PSD in the east (DAV) over solar cycle 24 (left *y*-axis) alongside F10.7 index (right *y*-axis) and (**c**,**d**) its dominance at low and high solar activity levels. Information in (**e**–**g**) is similar to (**b**–**d**) but for the west (LKW).

Next, PSD was seasonally dependent, as shown in Figure 9. PSD varied symmetrically according to the season in both parts of Southeast Asia. However, most variations disagreed with the F10.7 index, except for the east during Equinox, which was inversely correlated (R = −0.64). Compared to other seasons, the percentage occurrence of PSD during the Equinox was also among the highest. Therefore, this season was dominant, as seen in (b) and (d), except in the west at the high solar activity level, where its percentage was almost equivalent to J-solstice. Furthermore, depletion before dusk was observed, but the

occurrence was most remarkable at 05:00 LT and only represented 14% in the east and 12% in the west.

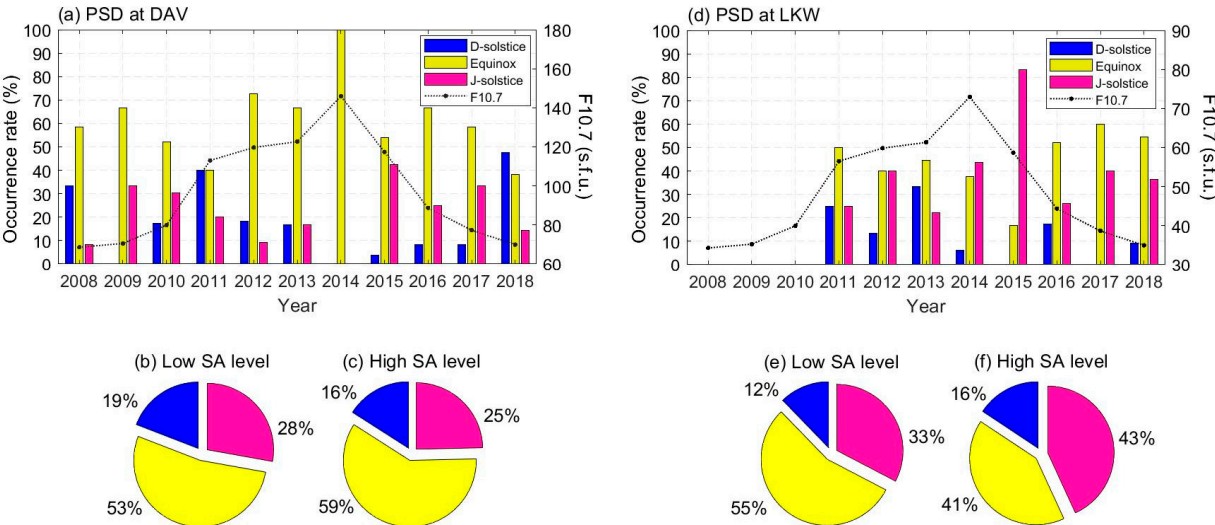

**Figure 9.** (**a**) Seasonal variation of PSD in the east (DAV) over solar cycle 24 (left *y*-axis) alongside F10.7 index (right *y*-axis), and (**b**,**c**) its dominance at low and high solar activity levels. Information in (**d**–**f**) is similar to (**a**–**c**) but for the west (LKW).

Previous studies reported a relation between evening CEJ and the growth of equatorial plasma bubbles (EPBs) in the ionospheric F-region. It was found that the development of EPBs around sunset was stunted when evening CEJ was present [34–37]. In other words, if EPBs existed, evening CEJ could not be observed. Even so, the EPB could still be generated under evening CEJ if the neutral wind blows eastward to generate the eastward electric field in the evening equator [38]. Thus, a further statistical study on the relation between evening CEJ (PSD) and EPB generation is needed, which had not previously been investigated because CEJ is a daytime phenomenon, and observation made was limited to 18:00 LT only.

## 4. Conclusions

This study observed CEJ over solar cycle 24 in eastern and western parts of Southeast Asia. It was apparent that MCEJ and ACEJ were solar activity dependent, with positive and negative correlations with the F10.7 index, respectively, in both parts, with remarkable symmetry. The asymmetricity of CEJ dominance (i.e., MCEJ in the east and ACEJ in the west) was probably due to local effects, which calls for a detailed study specifically in Southeast Asia to ascertain such effects in the sector, since the reported observation in the past was limited to only in the Indian sector. The seasonal variation of CEJ at different solar activity levels exhibit east–west symmetrical behavior during Equinox season for the greatest MCEJ occurrence and J-solstice at high solar activity level for ACEJ. This observation thus confirmed the mechanism that was proposed in past studies that involves atmospheric tides and stationary planetary waves. However, ACEJ at low solar activity level was excluded due to the corresponding percentage between Equinox and solstice month that might be caused by insufficient data. Upon extending the observation period to detect occurrences after dusk, it was found that PSD was solar activity independent in both parts and varied symmetrically by seasons. This event was also prominent during Equinox, except in the west during high solar activity level. In this preliminary PSD observation, we proposed that the appearance of PSD could be related to the equatorial plasma bubble phenomenon as the evening CEJ could have shifted to dusk time. This proposal will be confirmed in a future study on the CEJ–EPB relationship.

**Author Contributions:** Conceptualization, N.I.M.R. and N.S.A.H.; data curation, A.Y. and T.U.; funding acquisition, N.S.A.H. and M.A.; investigation, N.I.M.R.; methodology, N.I.M.R., N.S.A.H. and B.R.; project administration, N.S.A.H. and M.A.; supervision, N.S.A.H. and M.A.; validation, N.S.A.H. and K.A.Y.; visualization, N.I.M.R.; writing—original draft, N.I.M.R.; writing—review and editing, N.S.A.H., K.A.Y., T.U. and B.R. All authors have read and agreed to the published version of the manuscript.

**Funding:** This work was funded by the grant GUP-2019-053 from Universiti Kebangsaan Malaysia. MAGDAS was supported by the Japan Society for the Promotion of Science (JSPS) KAKENHI grant no. 268022. A.Y. was supported in part by the JSPS Core-to-Core Program (B. Asia-Africa Science Platforms), Formation of Preliminary Center for Capacity Building for Space Weather Research and JSPS KAKENHI grant 15H05815.

**Institutional Review Board Statement:** Not applicable.

**Informed Consent Statement:** Not applicable.

**Data Availability Statement:** Not applicable.

**Acknowledgments:** The authors would like to extend their gratitude to all members of the MAG-DAS project including MAGDAS Malaysia team and Malaysian Space Agency (MYSA) for their cooperation and contribution to this study. F10.7 data that were obtained from the GSFC/SPDF OMNIWeb interface.

**Conflicts of Interest:** The authors declare no conflict of interest.

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
