# Peer review of "The Variation of Counter-Electrojet Current at the Southeast Asian Sector during Different Solar Activity Levels"

_applsci, doi:10.3390/app12147138_

Round 1

Reviewer 1 Report

This study can be interesting to some readers. In the last round review, I provided some comments. However, the authors did not address well or even just ignore  them. I do not think this version can be accepted.

  1. There are still many abbreviations appeared in the text (MCEJ, ACEJ, PSD; SA), which causes distraction for the reader to understand the paper. When I read this paper, I cannot concentrate on reading it.
  2. I asked this issue as well: They suggest nonmigrating tides and stationary planetary waves to contribute to the observed east-west difference in CEJ. But they did not show any evidence to support this idea. How about the magnetic field line configuration effect to the east-west different in CEJ? There are many studies to report the zonal differences in both large scale (more than 1000 km) and small (100-300 km) scales in the ionosphere. However, this study did NOT cite those references.
  3. They just presented what they saw in the plots but without discussing the causes. 

Reviewer 2 Report

1) in line 22 what authors mean by EUEL?

2)±3 is it degree or what?

3) what does Kp reference in the line 83 , only as I am an experts in the article will be able to understand, and what about the new readers?

4) What does S24 authors are requested to convenience the article in a such a way a new readers can understand easily.

5) why only 2016, 2017, and 2018 are excluded? why not others justify?

6) Results and discussion very well discussed, as a reviewer  I don't suggest any changes

7) Lot of abbreviations and nomenclatures authors are requested to make a tabular and accepted after minor revisions 

Sincerely

Reviewer 

Reviewer 3 Report

The authors intend to investigate the longitudinal variation of counter electrojet (CEJ) in Southeast Asia. To make a better organization of the paper, major concerns are attached for authors to improve their manuscript. Therefore, potential readers can easily follow the content of the manuscript.

Round 2

Reviewer 1 Report

The authors addressed the comments I had. The paper can be accepted. 

Author Response

Thank you for the acceptance of our manuscript. 

Reviewer 3 Report

The authors have addressed my comments and questions and revised the manuscript based on my comments. For me, the revised manuscript is now better than the previous one. The current version of the manuscript can be published.